# Process-Induced Stress and Deformation of Variable-Stiffness Composite Cylinders During Curing

**DOI:** 10.3390/ma12020259

**Published:** 2019-01-14

**Authors:** Guiming Zhang, Jihui Wang, Aiqing Ni

**Affiliations:** 1School of materials science and engineering, Wuhan University of Technology, 122 Luoshi Road, Wuhan 430070, Hubei, China; zhangguiming20@whut.edu.cn (G.Z.); Jhwang@whut.edu.cn (J.W.); 2State Key Laboratory of Advanced Technology for Materials Synthesis and Processing, Wuhan University of Technology, 122 Luoshi Road, Wuhan 430070, Hubei, China

**Keywords:** composite materials, variable stiffness, cylinder, process-induced deformation, curing

## Abstract

Predicting and controlling process-induced deformation of composites during cure can play a significant role in ensuring the accuracy of manufacture and assembly of composite structures. In this paper the parametric investigation on the process-induced stress and deformation of variable-stiffness composite cylinders was presented. The Kamal model was used to simulate the cure kinetic for carbon/epoxy prepreg. A cure hardening instantaneously linear elastic (CHILE) constitutive model was adopted to determine the modulus of matrix resin. Self-consistent micro-mechanical models were employed to represent the mechanical properties and behaviors of the lamina. The three-dimensional model of a variable-stiffness composite cylinder was established using a linear fiber angle variation. The influence of the inner radius, the fiber end angle and the thickness on the stress and deformation of the variable-stiffness cylinder was evaluated using ABAQUS. The results show that the maximum stress increases with increases of the inner radius, the fiber end angle and the thickness. The inner radius of the cylinder have little effect on deformation, the deformation increases as the fiber end angle and the thickness increases. The present model and method can provide a useful tool for prediction of variable-stiffness composite cylinders.

## 1. Introduction

Composite structures have been widely used in aircrafts, marine, wind energy, and new energy automobiles owing to their superior mechanical advantages. The advantages of the traditional composite structures have not yet been completely exploited due to the constant stiffness throughout the layer made of straight fibers. Compared with the conventional straight-fiber laminates, using variable-stiffness laminates can tailor the stiffness distribution and improve the structural performance by means of fiber-angle altering. Process-induced deformation is inevitable during cure of composite structures. The deformation leads to a severe inaccuracy of the resulting shape and the assembly of the composite parts. Various factors such as the difference in coefficient of thermal expansion (CTE) between fiber and matrix, thermal strains of the part and the tool, heat transfer within the part and the tool and at their boundaries, the cure cycle, resin flow, chemical shrinkage of the matrix, the curing kinetics of the matrix, and the tool and part interaction will result in process-deformation of composite structures after demolding.

Various researchers have so far investigated the process-induced stress and deformation of non-planar parts. The behavior of thin non-planar laminates (L, T, U shapes) is not the same as thin flat panels; even though the structure is balanced and symmetrical, the conditions for obtaining stresses and deformations are much more complicated and cause geometric deviations [1]. Radford et al. [2] developed a method by which the stacking-sequence of the laminate was modified to counterbalance the environmentally induced shape accuracy. Ersoy et al. [3] implemented a two-step finite element model (FEM) to calculate the spring-back of C-shaped parts. Wisnom et al. [4] found that the spring-in angle of the laminate is proportional to the through-thickness strain between gelation and vitrification. Çınar et al. [5] found that fiber wrinkling at the corner sections of L-shaped parts decreases the spring-in angles. Johnston et al. [6] accurately calculated the spring-in angle of the final simple L-shaped components. Fernlund [7] proposed an engineering method for predicting the process-induced distortions of 3D composite parts. Dong et al. [8] studied the process-induced distortion of T-stiffened composite panels and found that the spring-in angle of the skin increases with the corner radius and the bonding length. Li et al. [9] investigated the distortion of a T-shaped part using a thermo-viscoelastic finite element analysis. Kappel [10] concentrated on the scattering of spring-in distortions of L-sectioned parts; the effects of various tooling materials and layups and the part thickness on spring-in of L-parts were also evaluated. 

Mezeix et al. [11] presented an original method to predict the deformation of composite flight structures using ABAQUS. The in-plane stresses and out-of-plane shear stress are the critical parameters for the warpage. Takagaki et al. [12] investigated the influences of the flange length, thickness, and shape on process-induced deformation. Chemical cure and shrinkages reduce the spring-in angles because of its orthotropic nature. Minakuchi et al. [13] utilized an optical fiber sensor to monitor the strain change in the corner of an L-shaped part throughout the structural life cycle. Bellini et al. [14] developed a thermo-chemical-mechanicalmodel to simulate the cure process and calculate spring-in angle of thin laminates.

Wiersma et al. [15] investigated three possible influences on process-induced deformation of the L-shaped component. Sun et al. [16] discussed the effects of tool-part interaction on L-shaped laminates during cure. The interactions between the L-shape laminate and the tool were simulated by introducing two shearing layers between the composite part and the tool. Roozbehjavan et al. [17] investigated the influence of stacking sequence on process-induced deformation of U-and L-shaped specimens. They found that the unbalanced stacking sequence panels resulted in large deformation than the asymmetric stacking-sequence panels. Baran et al. [18] developed a numerical simulation to compute the process-induced shape change in a pultruded L-shaped part. Liu et al. [19] proposed a hot-sizing process for forming composite materials to improve their shape accuracy.

Hao et al. [20] optimized the stress distribution of variable-stiffness plates with cutout based on flow field formula. Falcó et al. [21] analyzed the tow-drop effects on variable-stiffness panels using nonlinear finite element analyses. Liang et al. [22] investigated the nonlinear buckling behaviors of variable-stiffness panels with the reduced order model. Blom et al. [23] described the four different theoretical path along the length of conical structure. Tullu et al. [24] calculated the deformation and stress distributions of the variable-stiffness composite cylinder using ABAQUS. Rouhi et al. [25] analyzed buckling capacity of variable-stiffness elliptical composite cylinders based on design optimization method. Khani et al. [26] compared the variable-stiffness and the constant stiffness composite cylinders for maximum buckling capacity design.

Most previous researches have only been carried out on evaluation of process-induced deformations of straight-fiber composite laminates. However, few writers have been able to investigate the process-induced deformations of variable stiffness composites during cure. However, variable-stiffness composites made of curvilinear steered fiber paths have gained increasing attention and applications in many industries due to their excellent designability for achieving more design freedom and structural performance. Therefore, the investigation of process-induced stress and deformation of variable-stiffness composites became imperative for design and manufacture of this novel class of composite structures. An objective of this study was to calculate the process-induced stress and deformation of variable-stiffness cylinders. The 3D model of variable-stiffness cylinders was established. A parameterized investigation on process-induced stress and deformation of cylinders were carried out.

## 2. Basic Theory

### 2.1. Heat Transfer Equations

The model of the heat transfer is evaluated using the equation
(1)$$k_{x}\frac{\partial^{2}T}{\partial^{2}x}+k_{y}\frac{\partial^{2}T}{\partial^{2}y}+k_{z}\frac{\partial^{2}T}{\partial^{2}z}+Q=\rho C_{c}\frac{\partial T}{\partial t}$$
where *k_x_*, *k_y_*, and *k_z_* represent the thermal conductivities in the *x*, *y*, *z* directions, respectively; *T* is the kelvin temperature; *ρ* is the composite density; *C*_c_ stands for the specific heat of the laminates; *t* is time; *Q* is the instantaneous heat, given by
(2)$$Q=\rho_{r}\left(1-V_{f}\right)H_{r}\frac{d\alpha }{dt}$$
where *ρ_r_* is the resin density; *V_f_* represents the fiber volume fraction; *dα*/*dt* denotes the instantaneous curing rate; *H_r_* stands for the total thermal energy generated by reaction. The thermal parameters for Equations (1) and (2) are shown in Table 1.

### 2.2. Cure Kinetic Equations

The Kamal model was employed to express the epoxy resin cured reaction. The AS4/3501-6 prepreg was here used. The instantaneous curing rate for Hercules 3501-6 resin can be expressed as
(3)$$\frac{d\alpha }{dt}=\{\begin{array}{c} \left(K_{1}+K_{2}\alpha \right)\left(1-\alpha \right)\left(0.47-\alpha \right)\;\alpha \leq 0.3\\ K_{3}\left(1-\alpha \right)\;\;\;\;\;\;\;\;\alpha >0.3 \end{array}$$
where *K**_i_* (*i* = 1, 2, 3) are the rate constants, which is dependent of Arrhenius equation
(4)$$K_{i}=A_{i}\exp\left(\frac{-\Delta E_{i}}{RT}\right)\;\;\left(i=1,2,3\right)$$
where *A**_i_* (*i* = 1, 2, 3) denotes the pre-exponential constants; Δ*E**_i_* is the activation energy; *R* stands for the universal gas constant (*R* = 8.31 J/mol·K). The curing constants for Equation (4) are shown in Table 2.

### 2.3. Constitutive Model

The constitutive model of anisotropic composite materials can be expressed as,
(5)$$\left\{\begin{array}{c} \sigma_{1}\\ \begin{array}{l} \sigma_{2}\\ \sigma_{3}\\ \tau_{23}\\ \tau_{13} \end{array}\\ \tau_{12} \end{array}\right\}=\left[\begin{array}{cccccc} C_{11} & C_{12} & C_{13} & C_{14} & C_{15} & C_{16}\\ C_{21} & C_{22} & C_{23} & C_{24} & C_{25} & C_{26}\\ C_{31} & C_{32} & C_{33} & C_{34} & C_{35} & C_{36}\\ C_{41} & C_{42} & C_{43} & C_{44} & C_{45} & C_{46}\\ C_{51} & C_{52} & C_{53} & C_{54} & C_{55} & C_{56}\\ C_{61} & C_{62} & C_{63} & C_{64} & C_{65} & C_{66} \end{array}\right]\left\{\begin{array}{c} \varepsilon_{1}\\ \begin{array}{l} \varepsilon_{2}\\ \varepsilon_{3}\\ \gamma_{23}\\ \gamma_{13} \end{array}\\ \gamma_{12} \end{array}\right\}$$
where *C**_ij_* is the stiffness constants of the composite, given by.

(6)$$\left\{ \begin{array}{l} C_{11}=\frac{1-\upsilon_{23}\upsilon_{32}}{E_{2}E_{3}\Delta },\;C_{22}=\frac{1-\upsilon_{13}\upsilon_{31}}{E_{1}E_{3}\Delta },\;C_{33}=\frac{1-\upsilon_{12}\upsilon_{21}}{E_{2}E_{1}\Delta }\\ C_{12}=\frac{\upsilon_{12}+\upsilon_{13}\upsilon_{32}}{E_{2}E_{3}\Delta },\;\;C_{12}=\frac{\upsilon_{13}+\upsilon_{12}\upsilon_{23}}{E_{2}E_{3}\Delta },\;C_{23}=\frac{\upsilon_{23}+\upsilon_{21}\upsilon_{13}}{E_{1}E_{3}\Delta }\\ C_{44}=G_{12},\;C_{55}=G_{13},\;C_{66}=G_{23}\\ \Delta =\frac{1-\upsilon_{12}\upsilon_{21}-\upsilon_{13}\upsilon_{31}-\upsilon_{23}\upsilon_{32}-2\upsilon_{21}\upsilon_{32}\upsilon_{13}}{E_{1}E_{2}E_{3}} \end{array}\right.$$

The stress–strain relation of variable-stiffness composite plates can be written as
(7)$$\begin{array}{l} \left\{\begin{array}{c} \sigma_{x}\\ \sigma_{y}\\ \tau_{xy} \end{array}\right\}=\left[\begin{array}{ccc} \bar{C_{11}} & \bar{C_{12}} & \bar{C_{16}}\\ \bar{C_{21}} & \bar{C_{22}} & \bar{C_{26}}\\ \bar{C_{61}} & \bar{C_{62}} & \bar{C_{66}} \end{array}\right]\left\{\begin{array}{c} \varepsilon_{x}\\ \varepsilon_{y}\\ \gamma_{xy} \end{array}\right\}\\ \left\{\begin{array}{l} \tau_{yz}\\ \tau_{zx} \end{array}\right\}=\left[\begin{array}{cc} \bar{C_{44}} & \bar{C_{45}}\\ \bar{C_{45}} & \bar{C_{55}} \end{array}\right]\left\{\begin{array}{l} \gamma_{yz}\\ \gamma_{zx} \end{array}\right\} \end{array}\}$$
(8)$$\begin{array}{l} \bar{C_{11}}=U_{1}+U_{2}\cos2\theta (r)+U_{3}\cos4\theta (r)\\ \bar{C_{22}}=U_{1}-U_{2}\cos2\theta (r)+U_{3}\cos4\theta (r)\\ \bar{C_{12}}=U_{4}-U_{3}\cos4\theta (r)\\ \bar{C_{66}}=0.5(U_{1}-U_{4})-U_{3}\cos4\theta (r)\\ \bar{C_{16}}=0.5U_{2}\sin2\theta (r)+U_{3}\sin4\theta (r)\\ \bar{C_{26}}=0.5U_{2}\sin2\theta (r)-U_{3}\sin4\theta (r)\\ \bar{C_{44}}=U_{5}+U_{6}\cos2\theta (r)\\ \bar{C_{45}}=-U_{6}\sin2\theta (r)\\ \bar{C_{55}}=U_{5}-U_{6}\cos2\theta (r) \end{array}\}$$
in which *U**_i_* is the linear combination of positive axial moduli which is independent of the fiber angle *θ*, given by.

(9)$$\begin{array}{l} U_{1}=\frac{1}{8}(3C_{11}+3C_{22}+2C_{12}+4C_{66})\\ U_{2}=\frac{1}{2}(C_{11}-C_{22})\\ U_{3}=\frac{1}{8}(C_{11}+C_{22}-2C_{12}-4C_{66})\\ U_{4}=\frac{1}{8}(C_{11}+C_{22}+6C_{12}-4C_{66})\\ U_{5}=\frac{1}{2}\left(C_{44}+C_{55}\right)\\ U_{6}=\frac{1}{2}\left(C_{44}-C_{55}\right) \end{array}\}$$

### 2.4. Resin Modulus

The CHILE (α) model was here used to obtain the elastic modulus of the matrix resin. As shown in Figure 1, the changes in the resin modulus can be divided into three stages during cure: (a) the resin is in a viscous flow state; (b) the resin is violently cured and its elastic modulus significantly increases and the resin volume shrinks; (c) the resin is completely cured and no chemical reaction takes place. We assume that the elastic modulus of the resin is calculated from the gel point, which is recorded as *α* gel; when the resin reaches the glass state, the elastic modulus of the resin reaches a stable value, which is recorded as *α*_diff_. The instantaneous resin modulus can be written in terms of the degree of cure
(10)$$E=\{\begin{array}{cc} E_{r}, & \alpha <\alpha_{gel}\\ \left(1-\alpha_{\mathrm{mod}}\right)E_{r}+\alpha_{\mathrm{mod}}E_{g}+\gamma \alpha_{\mathrm{mod}}\left(1-\alpha_{\mathrm{mod}}\right)\left(E_{g}-E_{r}\right), & \alpha_{gel}<\alpha <\alpha_{diff}\\ E_{g}, & \alpha >\alpha_{diff} \end{array}$$
(11)$$\alpha_{mod}=\frac{\alpha -\alpha_{gel}}{\alpha_{diff}-\alpha_{gel}}$$
where *E* is instantaneous resin modulus; *E**_r_* and *E**_g_* are the incompletely cured and completely cured resin moduli, respectively; *α_gel_* and *α_diff_* stand for the boundary of degree of cure between the gel point and the glass state; *γ* describes the competitive mechanism between stress relaxation and chemical hardening. The modulus increases rapidly with the increase of *γ* at lower degrees of cure.

### 2.5. Chemical Shrinkage

Resin chemical shrinkage occurs when cure reaction occur and stops once the cure reaction is finished. Resin chemical shrinkage results in noticeable macroscopic deformation of composite parts. The strain contraction in all directions is assumed equal. Δ*ε*_r_ is the incremental isotropic shrinkage strain and Δ*v*_r_ is the incremental specific volume of resin shrinkage, resulting in Δ*ε*_r_, both of which can be related by

(12)$$\Delta \varepsilon_{r}=\sqrt[{3}]{1+\Delta v_{r}}-1$$

The incremental specific volume resin shrinkage can be defined as
(13)$$\Delta v_{r}=\Delta \alpha_{sh}\cdot v_{sh}$$
(14)$$\Delta \alpha_{sh}=\frac{\Delta \alpha }{\alpha_{diff}}$$
where Δα is the incremental degree of cure; *v*_sh_ is the total specific volume shrinkage of the completely cured resin, which is listed in Table 3.

### 2.6. Thermal Expansion Strain

The incremental strain in the longitudinal and transverse directions of a composite lamina can be estimated by
(15)$$\left\{ \begin{array}{l} \Delta \varepsilon_{1}^{th}=CTE_{1}\cdot \Delta T\\ \Delta \varepsilon_{2}^{th}=CTE_{2}\cdot \Delta T \end{array}\right.$$
where *CTE*_1_ and *CTE*_2_ are instantaneous effective *CTE*_s_ in the longitudinal and transverse directions, respectively; Δ*T* is the temperature increment within two consecutive time steps.

### 2.7. Micro-Mechanics Model

According to the self-consistent micro-mechanical model, the mechanical properties of the laminate can be calculated. The formulas for calculating the mechanical properties for homogeneous lamination refer to [29]. The mechanical properties for AS4 fiber and 3501-6 epoxy resin can be given in Table 3.

The longitudinal, transverse and shear elastic moduli can respectively be calculated by the formula

(16)$$\left\{ \begin{array}{c} E_{11}=E_{11f}V_{f}+E_{r}\left(1-V_{f}\right)+\left[\frac{4\left(\upsilon_{r}-\upsilon_{13f}^{2}\right)k_{f}k_{r}G_{r}\left(1-V_{f}\right)V_{f}}{\left(k_{f}+G_{r}\right)k_{r}+\left(k_{f}-k_{r}\right)G_{r}V_{f}}\right]\\ E_{22}=E_{33}=\frac{1}{\left(1/4k_{T}\right)+\left(1/4G_{23}\right)+\left(\upsilon_{12}^{2}/E_{11}\right)}\\ G_{12}=G_{13}=G_{r}\left[\frac{\left(G_{13f}+G_{r}\right)+\left(G_{13f}-G_{r}\right)V_{f}}{\left(G_{13f}+G_{r}\right)-\left(G_{13f}-G_{r}\right)V_{f}}\right]\\ G_{23}=\frac{G_{r}\left[k_{r}\left(G_{r}+G_{23f}\right)+2G_{23f}G_{r}+k_{r}\left(G_{23f}-G_{r}\right)V_{f}\right]}{k_{r}\left(G_{r}+G_{23f}\right)+2G_{23f}G_{r}-\left(k_{r}+2G_{r}\right)\left(G_{23f}-G_{r}\right)V_{f}} \end{array}\right.$$

The major and transverse Poisson’s ratios can respectively be given by

(17)$$\left\{ \begin{array}{l} \upsilon_{12}=\upsilon_{13}=\upsilon_{12f}V_{f}+\upsilon_{r}\left(1-V_{f}\right)+\left[\frac{\left(\upsilon_{r}-\upsilon_{12f}\right)\left(k_{r}-k_{f}\right)G_{r}\left(1-V_{f}\right)V_{f}}{\left(k_{f}+G_{r}\right)k_{r}+\left(k_{f}-k_{r}\right)G_{r}V_{f}}\right]\\ \upsilon_{23}=\frac{2E_{11}k_{T}-E_{11}E_{22}-4\upsilon_{13}^{2}k_{T}E_{22}}{2E_{11}k_{T}} \end{array}\right.$$

The shear modulus of fiber in the transverse direction can be expressed as

(18)$$G_{23f}=\frac{E_{33f}}{2\left(1+\upsilon_{23f}\right)}$$

The shear modulus of the resin can be determined by

(19)$$G_{r}=\frac{E_{r}}{2\left(1+\upsilon_{r}\right)}$$

The isotropic plane strain bulk modulus can be represented as

(20)$$k=\frac{E}{2\left(1-\upsilon -2\upsilon^{2}\right)}$$

The effective bulk modulus of the lamina can be computed by

(21)$$k_{T}=\frac{\left(k_{f}+G_{r}\right)k_{r}+\left(k_{f}-k_{r}\right)G_{r}V_{f}}{\left(k_{f}+G_{r}\right)-\left(k_{f}-k_{r}\right)V_{f}}$$

In the above equations, the subscripts *f* and *r* represent the fiber and resin; 1, 2, and 3 denote the three principal directions of the lamina, respectively.

### 2.8. Linear Fiber Angle Variation

The steered fiber paths with linear fiber angle variation were presented by Gürda et al. [32]. Assuming that the initial coordinate of the fiber path (*x*_0_, *y*_0_) is located at the center of the plate, the fiber angle *θ*(*r*), which is defined as the angle between the tangent of the fiber path and the x-axis, shows linear variation with x-coordinate, as illustrated in Figure 2.
(22)$$\theta \left(r\right)=\{\begin{array}{cc} \varphi +\frac{1}{d}\left(T_{0}-T_{1}\right)r+T_{0} & \;\;for\;-d\leq r\leq 0\\ \varphi +\frac{1}{d}\left(T_{1}-T_{0}\right)r+T_{0} & \;\;for\;\;\;0\leq r\leq d \end{array}$$
where *φ* represents the rotation angle; *T*_0_ stands for the initial fiber angle at the center of the plate; *T*_1_ denotes the fiber angle at the end; *d* is the length of the plate. Consequently, a reference fiber path can be expressed as <*T*_0_ǀ*T*_1_> and defines the variable-stiffness composite ply. ±<*T*_0_ǀ*T*_1_> represents the two adjacent plies, the fiber angles of which are equal and opposite at any *r*-coodinate. *φ*<*T*_0_ǀ*T*_1_> which varies linearly along *r*-direction stand for the linear fiber angle variation. The coordinate of the fiber reference path can be calculated by
(23)$$s\left(r\right)=\{\begin{array}{cc} \frac{d}{\left(T_{0}-T_{1}\right)}\left\{-\ln\left[\cos\left(\frac{1}{d}\left(T_{0}-T_{1}\right)r+T_{0}\right)\right]+C_{1}\right\} & \;\;for\;-d\leq r\leq 0\\ \frac{d}{\left(T_{1}-T_{0}\right)}\left\{-\ln\left[\cos\left(\frac{1}{d}\left(T_{1}-T_{0}\right)r+T_{0}\right)\right]+C_{2}\right\} & \;\;for\;\;\;0\leq r\leq d \end{array}$$
where *C*_1_, *C*_2_ denote the constants of integration, respectively.

### 2.9. Fiber Path Definitions on Cylinder

The location of a point on the fiber path of cylinder can be expressed in a curvilinear coordinate (*r*, *θ*, *x*). The *r*, *θ* and *x* are the radial, circumferential and axial axes, respectively. As shown in Figure 3, the original points of Cartesian coordinate (*z*, *y*, *x*) and cylindrical coordinate (*r*, *θ*, *x*) are located at the center of left end of the cylinder. The coordinate axes and variable fiber path on cylinder is schematized in Figure 3. The fiber path with different initial fiber angle and fiber end angle, simulating based on MATLAB, are shown in Figure 4. The geodesic path is the shortest path of two points on a surface.

The location of a given point with the Cartesian coordinate (*k*, *j*, *i*) is expressed by the position vector *r*_o_ as

(24)$$r_{o}\left(r,\theta ,x\right)=r\left\{\cos\left(\theta \right)k+\sin\left(\theta \right)j\right\}+xi$$

The basis vectors (*e_r_*, *e_θ_*, *e_x_*) in coordinate axes (*r*, *θ*, *x*) directions are derived with respect to Cartesian coordinate, respectively.

(25)$$\left\{ \begin{array}{l} e_{r}=\frac{\partial r_{o}}{\partial r}=\cos\left(\theta \right)k+\sin\left(\theta \right)j\\ e_{\theta }=\frac{\partial r_{o}}{\partial \theta }=r\left\{-\sin\left(\theta \right)k+\cos\left(\theta \right)j\right\}\\ e_{x}=\frac{\partial r_{o}}{\partial x}=i \end{array}\right.$$

## 3. Finite Element Analysis of Process-Induced Stress and Deformation

In order to investigate the stresses and deformation of the variable-stiffness cylinder with various parameters, the commercial FEA package ABAQUS was employed in this study. Eight-node three-dimensional elements are used. Figure 5 shows the variable-stiffness cylinder with a layup of 0°±<0°ǀ45°>_2S_. The red lines represent the fiber path.

The simulation process was divided into three steps: (a) the temperature and the cure degree were calculated using DISP, HETVAL, USDFLD subroutines at heat transfer step; (b) the results were used to simulate the residual stress distributions with UMAT, UEXPAN subroutines at the coupled temp-displacement step; (c) the resulting stresses and deformation were calculated through the change in boundary conditions. The mechanical properties of fiber and resin were incorporated into ABAQUS using subroutine UMAT. The manufacturer’s recommended cure cycle consists of two ramps and two dwells. The temperature rose from room temperature to 383 K in 45 min and kept constant for 1 h. Then, the temperature heats up to 453 K and retained unchanged about 2 h. Finally, the temperature cooled to room temperature in 80 min. The temperature load was applied at the inner and outer surfaces of the part using subroutine DISP. All other outer surfaces were set to adiabatic. A 3D transient heat-transfer analysis was first implemented using the heat transfer analysis in ABAQUS. Subroutine HETVAL was adopted to describe the exothermic cure reaction. The pressure 0.7 MPa was applied at the outer surface during cure and was removed after demolding. On the inner surface, the part can move freely in the axial direction, and no displacement was allowed in the radial direction. After demolding, the inner surface was non-restraint.

The cylinders will be manufactured with the shifted method using an automated fiber placement machines in our future paper. The FBG sensor will be used to detect the stress and deformation during curing.

## 4. Results and Discussions

The deformation of the 3D finite element model of the cylinder with the length *D* = 400 mm, the laminate thickness *t* = 2 mm, the inner radius *r*_i_ = 100 mm, a stacking sequence of 0°±<0°ǀ45°>_2S_, as shown in Figure 6. It is illustrated that the solidification deformations of the cylinder in the middle area reach the maximum value. The model expands mainly along the radial direction. However, the deformation along the axis is very small. Because the modulus along the axis is greater than that along radial direction, and the CTE along the axis is less than that along radial direction. This makes it easier to deform along the radial direction. A parametric model was established to investigate the influence of the inner radius, the fiber orientation, and the laminate thickness on the stress and deformation of variable-stiffness cylinders using ABAQUS.

### 4.1. Influence of the Inner Radius

FEM simulation was carried out to evaluate the influence of the inner radius ri on the stresses and deformation. The input parameters of the model are: the length of cylinder *D* = 400 mm, the laminate thickness *t* = 2 mm, the stacking sequence 0°±<0°ǀ45°>_2S_. The five inner radii 80 mm, 100 mm, 120 mm, 140 mm, and 160 mm were given respectively. Figure 7 presents the comparison of the simulated results. The stress distribution of these models are similar. The minimum stress distribution area increases with the increase of radius of the cylinder. The maximal stress appears in the both ends and the minimal stress appears in the middle of the cylinder. One should note that the fiber angle becomes smaller as its location approaches to the middle of the cylinder; as the result of this, the residual stress distribution changes with the variation in the fiber angle. The maximum stresses of the cylinders with inner radius 80 mm and 160 mm are respectively 1.524 MPa and 1.716 MPa. The results show that the maximum stress increases by 11.2% as the inner radius doubles; this indicates that the maximum stress improves with the increase in the inner radius of the cylinder. Figure 8 presents the comparison of the true strain. It is revealed that the inner radius of the cylinder have little effect on deformation. The maximum and the minimum deformations obtained using the above models are almost identical to each other. The maximum deformation appears in the both ends and the minimum deformation appears in the middle of the cylinder. Furthermore, the deformation distribution of these five models are all similar.

### 4.2. Influence of the Fiber Angle

The influence of the fiber angle on the stress and deformation were here studied. The other parameters are considered as: the length of cylinder *D* = 400 mm, the laminate thickness *t* = 2 mm, and the inner radius *r*_i_ = 100 mm. The finite element models with layups of 0°±<0°ǀ30°>_2S_, 0°±<0°ǀ45°>_2S_, 0°±<0°ǀ60°>_2S_, 0°±<0°ǀ75°>_2S_, and 0°±<0°ǀ90°>_2S_ respectively, are established.

The maximal stress increases with the increases of the fiber end angle. Furthermore, the stress distribution varied with different fiber end angle. The minimum stress region moves from the middle to both ends with the increases of fiber end angle. On the contrary, the maximum stress region moves from both ends to the middle with the increases of fiber end angle. The lamina properties in the fiber direction are dominated by the properties of the fiber. During the curing process, the deformation in the longitudinal direction can be neglected due to the high stiffness of fiber and low CTE, the deformation in the direction transverse direction depend on matrix resin. In addition, the main deformation is circumferential expansion. When the fiber angle tends to the circumferential direction, the stress and deformation decrease. This phenomenon is shown in the Figure 9 and Figure 10. The maximum stresses of the cylinders with the end fiber angles 30° and 90° are 1.126 MPa and 1.855 MPa, respectively. When the fiber end angle is larger than 60°, the deformation patterns become more complex.

### 4.3. Influence of the Cylinder Thickness

The influence of the cylinder thickness on the stress and deformation were here investigated. The assumed parameters of the model are: the length of cylinder *D* = 400 mm, the inner radius *r*_i_ = 100 mm, the stacking sequence of 0°±<0°ǀ45°>_2S_. The five thicknesses *t* = 2, 4, 6, 8, and 10 mm were given and the corresponding *r*_i_/*t* is 50, 25, 50/3, 12.5, and 10, respectively.

The results show that the stress and deformation increases as the laminate thickness increases. However, the thickness has little effect on the stress distribution. The stress of the cylinders with thicknesses 2 mm and 10 mm are 1.55 MPa and 2.67 MPa, respectively, as shown in Figure 11. The distribution of the temperature and residual stress in the thinner part are more uniform. However, the heat release is more intense in the middle of the thicker part. The heat cannot easily transmitted outside of the structure. Figure 12 shows the comparison of the true strain with different thickness. The strain caused by thermal and chemical contraction is mainly in the thickness direction. It is revealed that the thickness of the cylinder have little effect on deformation distribution. The maximal deformation appears in the both ends and the minimal deformation appears in the middle of the cylinder. Furthermore, the deformation distribution of these five models are all similar.

## 5. Conclusions

In this paper the novel methodology with the fully 3D thermomechanical model was presented to simulate the process-induced stress and deformation of variable-stiffness composite cylinders. The variation in resin modulus, Kamal model, cure shrinkage and CTEs of the composite were incorporated into the FEM-based model. The mechanical properties of the lamina were directly calculated using self-consistent micro-mechanical models. The FEM-based calculation for obtaining the process-induced deformation was performed in the three steps: (a) the temperature and the cure degree were calculated through the thermochemical module; (b) the residual stress distributions were computed through the thermomechanical module; and (c) the resulting deformation was calculated through demolding.

The parametric cylinders were modeled to investigate the effects of the inner radius, the fiber angle, and the laminate thickness on stress and deformation of the parts. The maximum stress increases by 11.2% as the inner radius doubles and it increases by 72.3% as the cylinder thickness increases from 2 mm to 10 mm. It should be noted that the maximum stress increases with increases of the inner radius, the fiber end angle and the thickness. The fiber end angle and the thickness of the cylinder have great effect on stress. The stress distribution and deformation patterns varied with different fiber end angle. It is also indicated that the inner radius of the cylinder has little effect on deformation. The thickness of the cylinder has great effect on maximum deformation. The deformation increases as the fiber end angle and the thickness increases.

## Figures and Tables

**Figure 1 materials-12-00259-f001:**
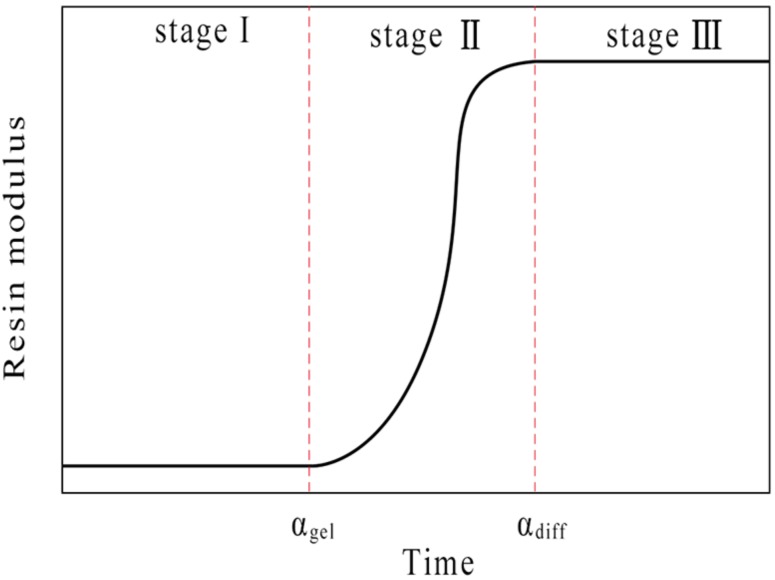
Resin modulus during cure.

**Figure 2 materials-12-00259-f002:**
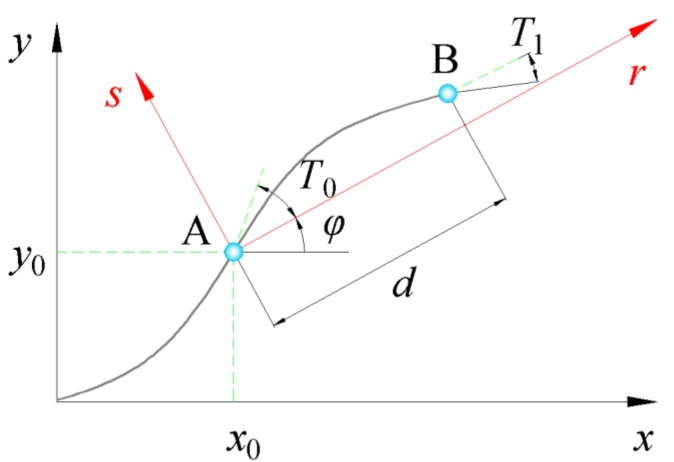
Fiber path with linear angle variation.

**Figure 3 materials-12-00259-f003:**
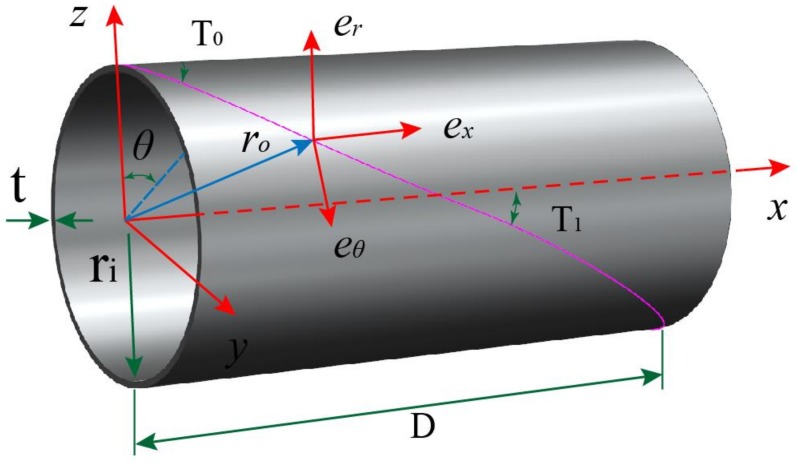
Coordinate axes and variable fiber path on cylinder.

**Figure 4 materials-12-00259-f004:**
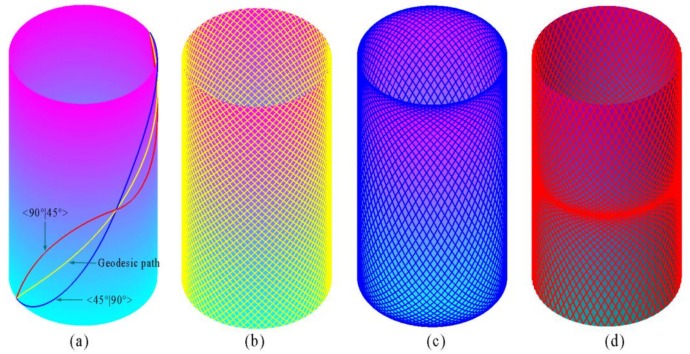
Simulation of fiber path with different initial fiber angle and end fiber angle on cylinder.

**Figure 5 materials-12-00259-f005:**
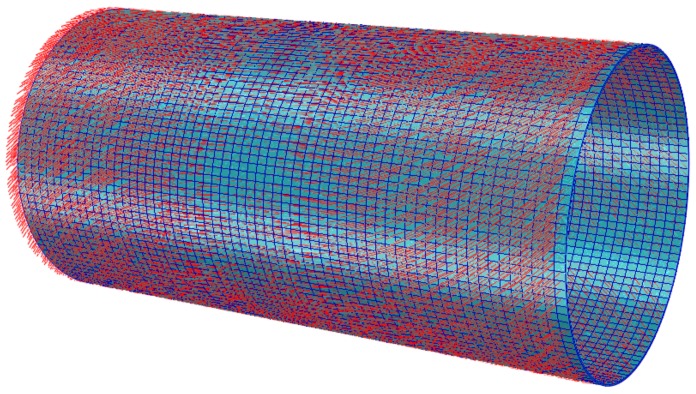
Finite element model of cylinder.

**Figure 6 materials-12-00259-f006:**
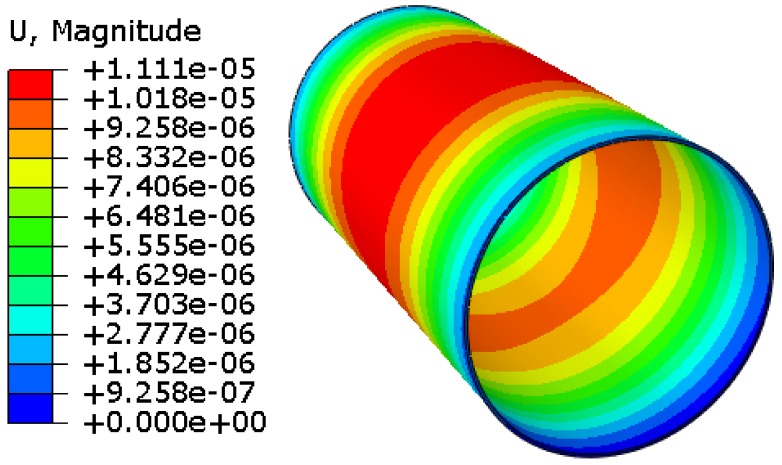
Deformation for the variable-stiffness cylinder.

**Figure 7 materials-12-00259-f007:**
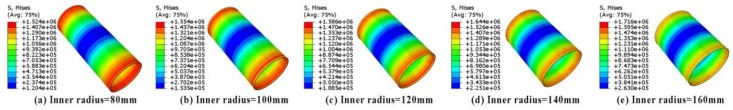
Influence of the inner radius on stress of variable-stiffness composite cylinders.

**Figure 8 materials-12-00259-f008:**
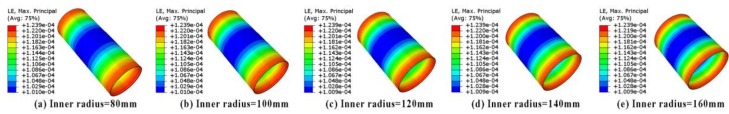
Influence of the inner radius on deformation of variable-stiffness composite cylinders.

**Figure 9 materials-12-00259-f009:**
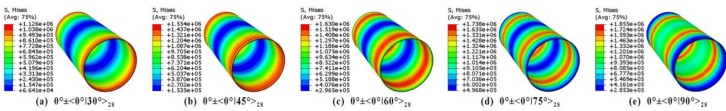
Influence of fiber angle on stress of variable-stiffness composite cylinders.

**Figure 10 materials-12-00259-f010:**
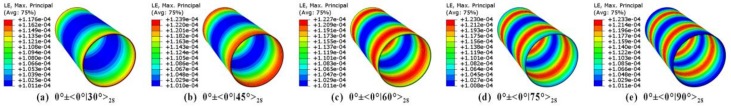
Influence of fiber angle on deformation of variable-stiffness composite cylinders.

**Figure 11 materials-12-00259-f011:**
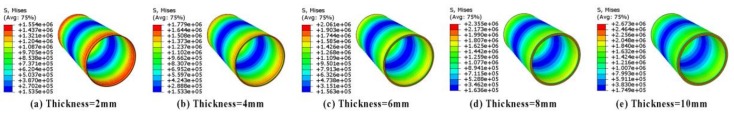
Influence of the thickness on stress of variable-stiffness composite cylinders.

**Figure 12 materials-12-00259-f012:**
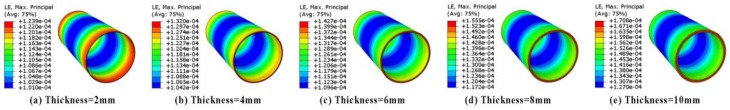
Influence of the thickness on deformation of variable-stiffness composite cylinders.

**Table 1 materials-12-00259-t001:** Thermal parameters for the AS4/3501-6 prepreg [27].

Constant	Value
*ρ*/kg·m^−3^	1578
*C_c_*/J·(kg·K)^−1^	862
*k_x_*/W·(m·K)^−1^	12.83
*k_y_* = *k_z_*/W·(m·K)^−1^	0.4135
*H_r_*/J·kg^−1^	198.9 × 10^3^

**Table 2 materials-12-00259-t002:** Curing constants for 3501-6 epoxy resin [28].

Constant	Value
*A*_1_/min^−1^	2.101 × 10^9^
*A*_2_/min^−1^	−2.014 × 10^9^
*A*_3_/min^−1^	1.960 × 10^5^
Δ*E*_1_/J·mol^−1^	8.07 × 10^4^
Δ*E*_2_/J·mol^−1^	7.78 × 10^4^
Δ*E*_3_/J·mol^−1^	5.66 × 10^4^

**Table 3 materials-12-00259-t003:** Mechanical properties for AS4 carbon fiber and 3501-6 resin [30,31].

Property	3501-6 Epoxy Resin	AS4 Fiber
*E*_1_/GPa	*E_g_*	210
*E*_2_/GPa	*E_g_*	17.2
*G*_12_ = *G*_13_/GPa	1.19	27.6
*G*_23_/GPa	1.19	27.6
υ_12_ = υ_13_	0.35	0.2
υ_23_	0.35	0.25
*CTE*_1_/(με/K)	57.6	−0.9
*CTE*_2_/(με/K)	57.6	7.2
*V_sh_*/%	2	--
*E_g_*/GPa	3.447	--
*E_r_*/GPa	3.447 × 10^−3^	--

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
