# Peer review of "Process-Induced Stress and Deformation of Variable-Stiffness Composite Cylinders During Curing"

_materials, 2019, doi:10.3390/ma12020259_

Reviewer 1 Report

1- kilograms should be indicated as 'kg' all over the paper

2- there is no connection between Eq. 5 and Eq. 6 and subsequent ones

3- after Eq. 22 'f' and 'r' should be italic as in the equations

4- Geodesic path in Figure 4 should be described

5- due to the thickness of the cylidner and the kind of model presented in Eq. 6 why 3d solid FE are used? Eq. 6and subsequent are typical of reduced elastic coefficients of shell elements. Please justify.

6- How many finite elements are employed along the cylinder thickness? This selection leads to different results in the simulations.

7- No comparison nor convergence simulation is presented. This leads to unsufficient presentation of the research.

8- Many recent research papers on variable stiffness shells have not been included in the introductory section.

All things considered the paper needs major revisions.

Major revisions:

- Description of numerical results totally insufficient.

- No comparison with the existing literature

- No convergence simulation of the FE modelling presented.

- Theoretical background not clear with reference to the numerical model solved.

Author Response

Comment 1:

kilograms should be indicated as 'kg' all over the paper.

Response to comments:

We are very sorry for our incorrect writing of the units. According to the reviewer’s suggestion, We have made correction. the units have been edited.

Comment 2:

there is no connection between Eq. 5 and Eq. 6 and subsequent ones.

Response to comments:

We have made revision in Eq. 5. wih the three dimensional of stress-strain relation.

Comment 3:

after Eq. 22 'f' and 'r' should be italic as in the equations.

Response to comments:

According to the reviewer’s suggestion, the 'f' and 'r' have been modified.

Comment 4:

Geodesic path in Figure 4 should be described.

Response to comments:

According to the reviewer’s suggestion, Geodesic path has been described.

Comment 5:

due to the thickness of the cylidner and the kind of model presented in Eq. 6 why 3d solid FE are used? Eq. 6and subsequent are typical of reduced elastic coefficients of shell elements. Please justify.

Response to comments:

We are very sorry for our incorrect using the formula. We have used the three dimensional of stress-strain relation in Eq. 5.

Comment 6:

How many finite elements are employed along the cylinder thickness? This selection leads to different results in the simulations.

Response to comments:

Due to the adoption of unsymmetrical stacking and in order to simplify the model, we only employ two elements in thickness. There are several layers in each element, depending on the thickness of the cylinder.

Comment 7:

No comparison nor convergence simulation is presented. This leads to unsufficient presentation of the research.

Response to comments:

This comment is very useful. We have made the convergence simulation in the early stage. We found the element size selected in this paper is suitable to analyze the model. Furthermore, the goal of this paper was to develop a novel methodology to predict process-induced residual stresses and deformation of variable-stiffness composite cylinders during cure and to evaluate the influential factors which reduce the cure-induced residual stress of the composite cylinder.

Comment 8:

Many recent research papers on variable stiffness shells have not been included in the introductory section.

Response to comments:

We are very sorry for our negligence of inclusion of recent papers on variable stiffness shells. We have added several related papers in the revised manuscript.

Comment 9:

Description of numerical results totally insufficient.

Response to comments:

We have modified the description of results in the revised manuscript according to reviewers suggestions.

Comment 10:

No comparison with the existing literature.

Response to comments:

We have carefully searched and checked some important published papers that are relevant with numerical model to simulate the development of degree of cure, residual stresses and deformations in the composite structures. In the previous research, the composite laminates with constant fiber angle have only been carried out on evaluation of process-induced stress and deformation. However, few variable stiffness composites(especial the variable stiffness composites cylinder) have been investigated on the process-induced stress and deformation during cure in the literature.

Comment 11:

No convergence simulation of the FE modelling presented.

Response to comments:

We have made the convergence simulation in the early stage. We found the element size selected in this paper is suitable to analyze the model. Furthermore, our goal is mainly focused on the effects of the parameter of cylinder to the stress and deformation.

Comment 12:

Theoretical background not clear with reference to the numerical model solved.

Response to comments:

    We are very sorry for our incorrect using the formula. We have used the three dimensional of stress-strain relation in Eq. 5, which correspond to3d solid FE.

Reviewer 2 Report

The topic of the manuscript is of interest and actuality.

The first part of the manuscript is well and clearly written. The description of the model is given in details and quite deep describes the problem.

The literature review is based on the appropriate number of sources and also quite well characterizes state of the art.

However, the second part of the manuscript (Results and discussion) lacks results of promised parametrical studies in the expected statement. Only qualitative descriptions of simulation results are given. No graphs are presented to illustrate the difference when inner radius, fiber angle or cylinder thickness are varied (if any). Also, the results obtained are poorly discussed.

The conclusion is just a summary of conducted investigations. No generalization or numerical data is provided in order to characterize the new achieved results. Also, the last paragraph of the conclusion mostly postulates that no substantial influence of parameters under study was revealed. In this case, what was the main idea of the study besides developing the model?

It is suggested to reconsider the manuscript after major revision.

Author Response

Comment 1:

The topic of the manuscript is of interest and actuality.

The first part of the manuscript is well and clearly written. The description of the model is given in details and quite deep describes the problem.

The literature review is based on the appropriate number of sources and also quite well characterizes state of the art.

Response to comments:

We greatly appreciate the reviewer’s efforts to carefully review the paper and the valuable suggestions offered. We are grateful for your good overall impression of the paper.

Comment 2:

However, the second part of the manuscript (Results and discussion) lacks results of promised parametrical studies in the expected statement. Only qualitative descriptions of simulation results are given. No graphs are presented to illustrate the difference when inner radius, fiber angle or cylinder thickness are varied (if any). Also, the results obtained are poorly discussed.

Response to comments:

We would like to gratefully thank the reviewers for their detailed feedback to improve our submission. We have added, deleted and rewritten some sentences from results and discussion according to reviewers suggestions. Based on the calculation results, we compared the results of the figures, and found some trend phenomena. We added some description of the representative phenomena with data.

Comment 3:

The conclusion is just a summary of conducted investigations. No generalization or numerical data is provided in order to characterize the new achieved results. Also, the last paragraph of the conclusion mostly postulates that no substantial influence of parameters under study was revealed. In this case, what was the main idea of the study besides developing the model?

Response to comments:

We greatly appreciate the reviewer’s efforts to carefully review the paper and the valuable suggestions offered. We have modified the description of results in the second paragraph of the conclusions according to reviewers suggestions.

Reviewer 3 Report

The authors explained the stresses and deformations developed in composite cylinders using a finite element (FE) model with ABAQUS. The literature review and the mathematical equations supporting this research looks good. The main thing lacking in this work is that the FE model is not supported by any experimental data. It is necessary for authors to fabricate several composite cylinders with different thicknesses and inner and outer radius, and verify the soundness of their results with experimental data.  

Author Response

Comment:

The authors explained the stresses and deformations developed in composite cylinders using a finite element (FE) model with ABAQUS. The literature review and the mathematical equations supporting this research looks good. The main thing lacking in this work is that the FE model is not supported by any experimental data. It is necessary for authors to fabricate several composite cylinders with different thicknesses and inner and outer radius, and verify the soundness of their results with experimental data.

Response to comments:

We would like to gratefully thank the reviewers for their detailed feedback to improve our submission. The experiment of the variable-stiffness composite cylinder will be published in our later articles. We have compared the experiment and the simulation of the composite plate. The experiment and the simulation layup of 454/-4516 is shown in the following enclosure. The calculation is consistent with experiment. The results show that the method of simulation which we used is appropriate to predict the stress and deformation.

Round  2

Reviewer 1 Report

Good comments have been provided by the authors.

However to avoid confusion, Eq. 5 which is related to 3D elastic materials and the following ones, should use C symbol not Q. Because Q is classically used for reduced elastic coefficients. On the contrary, C is world-wide used for 3D elastic constitutive material law.

Two finite elements along the thickness is generally not a good choice because so modeling does not represent the shear behavior of physical structures. In fact, in the literature 3 to 4 finite elements are employed though-the-thickness. Thus, more specific comments related to this aspect should be given.

The authors should be more specific on the given statement "There are several layers in each element, depending on the thickness of the cylinder." because what does it mean that several layers are on each element with 3D FE modelling only 1 single material configuration can be considered per "layer of elements" 3D FE brick elements cannot consider lamination schemes.

Author Response

Response to comments of Reviewer

Comment 1:

Good comments have been provided by the authors. However to avoid confusion, Eq. 5 which is related to 3D elastic materials and the following ones, should use C symbol not Q. Because Q is classically used for reduced elastic coefficients. On the contrary, C is world-wide used for 3D elastic constitutive material law.

Response to comments:

We greatly appreciate the reviewer’s efforts to carefully review the paper and the valuable suggestions offered. According to the reviewer’s suggestion, we have made correction. the C symbol have been replaced by Q.

Comment 2:

Two finite elements along the thickness is generally not a good choice because so modeling does not represent the shear behavior of physical structures. In fact, in the literature 3 to 4 finite elements are employed though-the-thickness. Thus, more specific comments related to this aspect should be given.

Response to comments:

Firstly, the eight-node biquadratic quadrilateral generalised plane strain elements are used in Ersoy’s research. Secondly, the dimension of the model in Ersoy’s research is smaller than our model. Finally, Ersoy investigated the constant stiffness structure and we investigated the variable-stiffness structure. Furthermore, the more elements will make the variable-stiffness model very complex.

Comment 3:

The authors should be more specific on the given statement "There are several layers in each element, depending on the thickness of the cylinder." because what does it mean that several layers are on each element with 3D FE modelling only 1 single material configuration can be considered per "layer of elements" 3D FE brick elements cannot consider lamination schemes.

Response to comments:

The fiber angle was assigned to every element using python script. And every element contains several layers and has different angle individually. The key script as follows.

compositeLayup.CompositePly(suppressed=False, plyName='Ply-1', region=region1,

material='Resin3501-CHILE', thicknessType=SPECIFY_THICKNESS, thickness=1.0,

orientationType=ANGLE_0, additionalRotationType=ROTATION_NONE,

additionalRotationField='', axis=AXIS_3, angle=0.0, numIntPoints=3)

compositeLayup.CompositePly(suppressed=False, plyName='Ply-2', region=region2,

material='Resin3501-CHILE', thicknessType=SPECIFY_THICKNESS, thickness=1.0,

orientationType=ANGLE_0, additionalRotationType=ROTATION_NONE,

additionalRotationField='', axis=AXIS_3, angle=0.0, numIntPoints=3)

p = mdb.models['Model-1'].parts['partname-mesh-1']

    set="Set-"

    for i in range(1,32):

        name= set+str(i)

        region = p.sets[name]

        p = mdb.models['Model-1'].parts['partname-mesh-1']

        f = p.elements

Reviewer 2 Report

Some corrections were made in the manuscript. Most of them are rather decorative. However, they are not grammatically correct. For instance:

Page 12, Line 261 "The inner radius double and maximum stress increase 11.2%. The maximum stress increases 72.3%". Please check and correct.

Page 12, line 265 "...of these models are almost same." Please check and correct.

Page 12, lines 285-286 "The maximum stress of the cylinders with fiber end angle 30° and 90° are 1.126 MPa and 1.855 MPa." Please check and correct.

Page 13, line 325. In the newly inserted text the grammar is incorrect: "The inner radius double and maximum stress increase 11.2%. The maximum stress increases 72.3%". Please check and correct.

Some numbers were added to the conclusion, However, there is still no generalizing statement. What was the general idea of the research? What is the advantage of the model overs others? Are the results more precise in contrast with literature? What are the prospects of the study? In a current form, the Conclusion looks like a summary. So, the generalization is required.

Author Response

Response to comments of Reviewer

Comment 1:

Some corrections were made in the manuscript. Most of them are rather decorative. However, they are not grammatically correct. For instance:

Page 12, Line 261 "The inner radius double and maximum stress increase 11.2%. The maximum stress increases 72.3%". Please check and correct.

Page 12, line 265 "...of these models are almost same." Please check and correct.

Page 12, lines 285-286 "The maximum stress of the cylinders with fiber end angle 30° and 90° are 1.126 MPa and 1.855 MPa." Please check and correct.

Page 13, line 325. In the newly inserted text the grammar is incorrect: "The inner radius double and maximum stress increase 11.2%. The maximum stress increases 72.3%". Please check and correct.

Response to comments:

We fully appreciate the reviewer’s concerns and suggestions. We are encouraged by the reassurance regarding the general quality of English by the reviewers. Consequently, we have modified the manuscript to improve the English writing. Furthermore, we have invited an associate professor, who returns from studying abroad in the United Kingdom and is a proficient English speaker, to go through the manuscript and improved the language.

Comment 2:

Some numbers were added to the conclusion, However, there is still no generalizing statement. What was the general idea of the research? What is the advantage of the model overs others? Are the results more precise in contrast with literature? What are the prospects of the study? In a current form, the Conclusion looks like a summary. So, the generalization is required.

Response to comments:

We have added the generalizing statement in the revised manuscript according to reviewers suggestions. We have carefully searched and checked some important published papers that are relevant with numerical model to simulate the development of degree of cure, residual stresses and deformations in the composite structures. In the previous research, the composite laminates with constant fiber angle have only been carried out on evaluation of process-induced stress and deformation. However, few variable stiffness composites(especial the variable stiffness composites cylinder) have been investigated on the process-induced stress and deformation during cure in the literature.

The objective of this paper was to develop a novel methodology to predict process-induced residual stresses and deformation of variable-stiffness composite cylinders during cure and to evaluate the influential factors which reduce the cure-induced residual stress of the composite cylinder.

Reviewer 3 Report

The authors should write a paragraph about the experiment they are doing to confirm the soundness of their FE model and refer the readers to their future paper for more details.

Author Response

Response to comments of Reviewer

Comment:

The authors should write a paragraph about the experiment they are doing to confirm the soundness of their FE model and refer the readers to their future paper for more details.

Response to comments:

We greatly appreciate the reviewer’s  valuable suggestions offered. According to the reviewer’s suggestion, we have added a paragraph in the revised paper.